# Importance of Melatonin in Assisted Reproductive Technology and Ovarian Aging

**DOI:** 10.3390/ijms21031135

**Published:** 2020-02-08

**Authors:** Hiroshi Tamura, Mai Jozaki, Manabu Tanabe, Yuichiro Shirafuta, Yumiko Mihara, Masahiro Shinagawa, Isao Tamura, Ryo Maekawa, Shun Sato, Toshiaki Taketani, Akihisa Takasaki, Russel J. Reiter, Norihiro Sugino

**Affiliations:** 1Department of Obstetrics and Gynecology, Yamaguchi University Graduate School of Medicine, Minamikogushi 1-1-1, Ube 755-8505, Japan; mkawa@yamaguchi-u.ac.jp (M.J.); yshirafu@yamaguchi-u.ac.jp (Y.S.); andy-yu@yamaguchi-u.ac.jp (Y.M.); mshina@yamaguchi-u.ac.jp (M.S.); isao@yamaguchi-u.ac.jp (I.T.); rmaekawa@yamaguchi-u.ac.jp (R.M.); shnstymg@yamaguchi-u.ac.jp (S.S.); taketani@yamaguchi-u.ac.jp (T.T.); sugino@yamaguchi-u.ac.jp (N.S.); 2Department of Obstetrics and Gynecology, Saiseikai Shimonoseki General Hospital, Yasuokacho 8-5-1, Shimonoseki 759-6603, Japan; my071123my@yahoo.co.jp (M.T.); a-takasaki@simo.saiseikai.or.jp (A.T.); 3Department of Cellular and Structural Biology, The University of Texas Health Science Center, San Antonio, TX 78229, USA; reiter@uthscsa.edu

**Keywords:** melatonin, ovarian aging, reactive oxygen, oxidative stress, infertility

## Abstract

Melatonin is probably produced in all cells but is only secreted by the pineal gland. The pineal secretion of melatonin is determined by the light–dark cycle, and it is only released at night. Melatonin regulates biological rhythms via its receptors located in the suprachiasmatic nuclei of the hypothalamus. Melatonin also has strong antioxidant activities to scavenge free radicals such as reactive oxygen species (ROS). The direct free radical scavenging actions are receptor independent. ROS play an important role in reproductive function including in the ovulatory process. However, excessive ROS can also have an adverse effect on oocytes because of oxidative stress, thereby causing infertility. It is becoming clear that melatonin is located in the ovarian follicular fluid and in the oocytes themselves, which protects these cells from oxidative damage as well as having other beneficial actions in oocyte maturation, fertilization, and embryo development. Trials on humans have investigated the improvement of outcomes of assisted reproductive technology (ART), such as in vitro fertilization and embryo transfer (IVF-ET), by way of administering melatonin to patients suffering from infertility. In addition, clinical research has examined melatonin as an anti-aging molecule via its antioxidative actions, and its relationship with the aging diseases, e.g., Alzheimer’s and Parkinson’s disease, is also underway. Melatonin may also reduce ovarian aging, which is a major issue in assisted reproductive technology. This review explains the relationship between melatonin and human reproductive function, as well as the clinical applications expected to improve the outcomes of assisted reproductive technology such as IVF, while also discussing possibilities for melatonin in preventing ovarian aging.

## 1. Introduction

Melatonin is an indoleamine (molecular weight 232.3) produced by all cells including those in the pineal gland, located in the posterior wall of the third ventricle [1]. Melatonin is rhythmically released by the pineal gland such that it is mostly secreted during the night. This rhythm is controlled by the light–dark cycle, with light inhibiting melatonin synthesis and release. The circadian cycle of melatonin plays major roles in the regulation of body temperature, the secretion of various reproductively active hormones, and the circadian sleep–wake cycle [2]. Melatonin, which is both fat-soluble and water-soluble, is distinctive in that it can easily pass through cell membranes. It is not only present in the blood but also in body fluids, including cerebrospinal fluid, follicular fluid, and seminal vesicle fluid [3]. Melatonin receptors are found in numerous organs, and in addition to their own biorhythms, they determine, via several signaling mechanisms, secretion of various hormones, immune functions, lipid and glucose metabolism, as well as bone metabolism [4,5,6,7,8,9]. These rhythms are involved in aging, carcinogenesis, and a number of diseases [4,10,11,12,13,14,15,16]. Clinical research is underway in various fields including the application of melatonin in treating biological rhythm abnormalities, glucose and lipid metabolism disorders, immunoregulation, anticancer effects and aging-related diseases (Parkinson’s and Alzheimer’s disease) [17,18,19,20,21]. 

In addition to its receptor-mediated actions, melatonin’s ability to act as a direct free radical scavenger and as an indirect antioxidant has greatly broadened understanding of the mechanisms by which melatonin benefits reproductive physiology. Melatonin has potent antioxidative capacity to remove both reactive oxygen species (ROS) and reactive nitrogen species (RNS) [22]. As already noted, the scavenging actions are a direct effect and require no receptor mediation.

## 2. Antioxidative Effects of Melatonin

In 1993, the discovery of melatonin as an antioxidant was achieved [23,24]. Melatonin is a direct free radical scavenger with more powerful antioxidant potential than conventional antioxidants such as vitamin C and E, mannitol, and glutathione [25]. Melatonin is both fat and water soluble, and can pass easily through cell membranes [26]. Therefore, it exists in the cytosol as well as in mitochondria and nuclei [27,28,29,30]. Melatonin, through direct scavenging and indirect antioxidant actions, limits oxidative stress in all cells and protects DNA and other components from damages [31,32]. The antioxidative actions of melatonin and its metabolites are extremely vast, including the ability to neutralize superoxide anion (O_2_^•−^), hydroxyl radical (^•^OH), single oxygen (^1^O_2_), hydrogen peroxide (H_2_O_2_), hypochlorous acid (HOCl), nitric oxide (NO), and peroxynitrite anion (ONOO^−^) [33]. Melatonin is a valuable substance that particularly detoxifies the hydroxyl radical (^•^OH), which is highly reactive and toxic. When melatonin eliminates free radicals, it is converted to metabolites including cyclic 3-hydroxmelatonin (C3OHM), *N*1-acetyl-*N*2-formyl-5-methoxykynuramine (AFMK) and *N*1-acetyl-5-methoxykynuramine (AMK). These metabolites, like melatonin, have potent antioxidative actions [34]. Melatonin also indirectly increases antioxidant enzyme activity including superoxide dismutase (SOD) and glutathione peroxidase (GPx), and increases both antioxidant enzyme activity and mRNA expression via the membrane receptors of melatonin (MT1, MT2) [35]. 

## 3. Reactive Oxygen and Reproductive Function

ROS have important roles in reproductive processes such as influencing follicular growth, oocyte maturation, ovulation, fertilization, embryo implantation, and embryo development [36]. During ovulation, following a surge of luteinizing hormone (LH) that induces the release of ova, large amounts of ROS are generated by vascular endothelial cells and macrophages as neovascularization progresses within the follicles. The ROS generated within follicles provides the stimulation needed for oocyte maturation and follicular rupture. Excessive ROS, however, cause cytotoxicity. As a defense mechanism against ROS, there are antioxidative enzymes and antioxidants present in follicles and in the oocytes themselves [37], which protect oocytes and granulosa cells from oxidative stress. If an imbalance between ROS concentrations and antioxidant activities occurs, oocytes and granulosa cells are readily damaged due to oxidative stress, which results in poor oocyte quality. Studies have reported that in mice after the induction of ovulation with human chorionic gonadotropin (hCG) following stimulation of superovulation using pregnant mare serum gonadotropin (PMSG), increased concentrations of 8-hydroxy-2′-deoxyguanosine (8-OHdG), a marker of DNA damage, and of hexanoyl-lysine (HEL), a marker of lipid peroxidation, prior to ovulation were measured [38]. Also, in humans, it has been reported that patients undergoing in vitro fertilization (IVF) with a high rate of denatured oocytes (poor oocyte quality) exhibit a high levels of 8-OHdG in the follicular fluid; also, low fertilization rates were noted for oocytes retrieved from follicles with an elevated concentration of 8-OHdG in the follicular fluid [39]. Thus, it appears that ROS generated during the ovulatory process is a source of oxidative stress to oocytes, in follicles, as well as granulosa cells, providing an explanation for reduced oocyte quality and infertility.

## 4. Melatonin in the Ovaries

It is becoming clear that melatonin acts directly on the ovaries [40]. In a report that examined tissue penetration of melatonin in cats, it was found that, compared to other organs, the accumulation was highest in the ovaries [41]. Melatonin also is found at a high concentrations in human follicular fluid, and it increases in proportion to follicular growth [42]. Hence, there is a mechanism whereby melatonin is taken up into the follicles as they enlarge. The physiological significance as to why melatonin is taken up into follicles and is present at high concentrations in follicles has long been unclear. Since ovulation is perceived as a phenomenon that resembles inflammation, and as a defense mechanism against ROS generated in follicles, antioxidative enzymes and antioxidants, including melatonin, protect oocytes and granulosa cells from oxidative damage. It is highly likely that melatonin carries out major roles as an antioxidant in both follicles and oocytes. In a study of follicular fluid collected during oocyte retrieval from women who have undergone IVF-ET, there were no marked correlations observed between the concentration of 8-OHdG and antioxidative enzymes, including copper (Cu) and zinc (Zn) superoxide dismutase (Cu, Zn-SOD), as well as for glutathione; however, a significantly negative correlation was observed between the concentration of melatonin and 8-OHdG [39]. Furthermore, in patients who received melatonin (3 mg/day, taken at 22:00), the levels of melatonin in follicular fluid significantly increased compared to the period when melatonin was not given as a supplement, and 8-OHdG concentrations were significantly decreased as a result of melatonin administration [39]. These results suggest that melatonin reduces oxidative stress in ovarian follicles via its antioxidative action, thereby protecting oocytes and granulosa cells.

Tanabe et al. reported that melatonin protects granulosa cells from ROS by reducing oxidative stress of cellular components including nucleus, mitochondria, and cell membranes [43]. Mouse granulosa cells were incubated with H_2_O_2_ (0.1–10 mM) in the presence or absence of melatonin (100 μg/mL). DNA damage (8-OHdG and γH2AX), mitochondrial dysfunction, and lipid peroxidation of membranes (HEL) were elevated after H_2_O_2_ treatment, whereas these harmful effects of ROS on cellular components were alleviated by melatonin supplementation. Moreover, H_2_O_2_ treatment increased the number of apoptotic cells and caspase 3/7 (Csp3/7) activities, which were inhibited upon melatonin administration. These results document that while ROS damage DNA, mitochondria, and cell membranes of granulosa cells, melatonin prevents this mutilation, thereby protecting granulosa cells (Figure 1).

Tamura et al. examined the effect of ROS and melatonin on the maturation process of oocytes from mice [39]. Germinal vesicle (GV)-stage oocytes were cultured with H_2_O_2_. After 12 h incubation, oocytes with the first polar body (MII stage oocytes) were significantly decreased as a result of the addition of H_2_O_2_ in a dose-dependent manner (>200 μM). However, melatonin treatment dose-dependently blocked the inhibitory effect of H_2_O_2_ on oocyte maturation. To further investigate the intracellular role of melatonin, oocytes were incubated with dichlorofluorescein (DCF-DA), a fluorescent dye that identifies ROS [38]. High fluorescence intensities were observed in the presence of H_2_O_2_ (300 μM), whereas the increased fluorescence intensity was significantly depressed by melatonin treatment. This is consistent with antioxidative action of melatonin, which reduces ROS in oocytes. 

Based on these findings, it appears that melatonin, both locally-synthesized and secreted into the blood by the pineal gland, is taken up by ovarian follicles, where it locally reduces ROS in the follicles and limits oxidative stress, thereby protecting oocytes, as well as granulosa cells, and contributes to oocyte maturation and the luteinization of granulosa cells (Figure 1).

## 5. The Clinical Application of Melatonin in the Field of Reproductive Medicine

In recent years, remarkable advances have been made in assisted reproductive technology (ART) such as IVF-ET for infertility treatment; however, satisfactory conception rates have not been achieved. The major cause of this is attributed to problems in the quality of oocytes [44,45]. In IVF-ET, the frequent unsuccessful cases in which fertilization, embryo development, and implantation do not progress optimally are a result of poor oocyte quality. While the processes underlying reduced oocyte quality have not been fully elucidated, it is thought that oxidative stress caused by ROS in the follicle is a critical factor. In IVF-ET program, in vitro incubation including oocyte incubation, insemination (co-incubation with spermatozoa), fertilization, embryo development, and embryo transfer are performed in a high oxygen environment compared with the in vivo physiological condition. As long-term in vitro incubation conditions have a major negative impact on the quality of the oocyte and embryo, extreme caution must be exercised to control oxidative stress caused by ROS during incubation.

As an application of the antioxidative effects of melatonin for reproductive medicine, it is possible that in vivo administration may improve the quality of oocytes until ovulation (oocyte retrieval) and improve oocyte maturation, fertilization, and embryo development during in vitro incubation when melatonin is added to the incubation medium (Figure 2).

### 5.1. Melatonin in Assisted Reproductive Technology (ART)

Follicular melatonin protects granulosa cells and oocytes from ROS. If follicular concentrations of melatonin were increased by administering melatonin to female patients, it may well improve the quality of oocytes. Tamura et al. performed an examination of IVF-ET with infertile patients with poor oocyte quality (fertilization rate < 50%) who were divided into two groups, including a treatment group given melatonin tablets (3 mg/day) for one month up to the time of oocyte retrieval for the next IVF-ET (melatonin group) and a group without melatonin treatment (control group). In the melatonin treated subjects, the fertilization rate was approximately 50% (vs. approximately 20% for the control group) and the pregnancy rate was roughly 20% (vs. approximately 10% for the control group), indicating improved outcomes for IVF-ET [39]. It is possible that the administration of melatonin to patients suffering from infertility increases follicular concentrations of melatonin, which thereby inhibits oxidative stress and improves the quality of oocytes, thus improving fertilization and pregnancy rates. After this initial study, other reports also examined whether melatonin therapy improves the clinical outcome of IVF-ET, with the findings likewise also showing that the concurrent use of melatonin increases the number of mature oocytes, the fertilization rate, and number of high-quality embryos [39,46,47,48,49,50] (Table 1). It is suggested that the effects of melatonin include an increased concentration of melatonin in the follicular fluid and a reduced 8-OHdG concentration, which leads to the conclusion that melatonin’s multiple antioxidant actions lower oxidative stress in oocytes [39,50]. In ART, melatonin would be expected to elevate the pregnancy rate by improving the quality of oocytes as well as promoting the fertilization rate and embryo development.

### 5.2. Oocyte Maturation, Embryo Development, and Melatonin

In studies using oocytes from mice, cows, and pigs, it has been reported that oocyte maturation, fertilization, and embryo development are promoted upon in vitro incubation of oocytes in culture medium supplemented with melatonin (Table 2). As oxygen concentrations are higher in in vitro incubations than in vivo, oxidative stress caused by ROS generated during incubation often has an adverse effect on oocyte maturation and embryo development. The addition melatonin to the culture medium wound detoxifies ROS, thereby reducing oxidative damage so as to protect the oocytes and granulosa cells. Reducing oxidative stress and apoptosis of oocytes and promoting mitochondrial function via melatonin treatment have been shown to improve oocyte maturation, fertilization rate, and rate of blastocyst formation (blastocyst cell count) [51,52,53]. Furthermore, the effect of oxidative stress induced by substances that generate ROS, such as bisphenol A (BPA) and aflatoxin B1 (AFB1), is reduced when melatonin is added to the culture medium [54,55]. This action of melatonin reduces ROS and oxidative stress due to its direct antioxidative actions.

Because melatonin reduces ROS and oxidative stress via its direct antioxidative functions, it improves cellular integrity (Figure 3). However, indirect effects of melatonin via membrane receptors (MT1, MT2) and/or nuclear receptors (RORα) are also important to understand the processes by which melatonin alters the level of oxidative stress [56,57]. Figure 3 summarizes the mechanism of melatonin that are assumed to improve oocyte maturation and quality based on reports to date. Melatonin membrane receptors (MT1, MT2) are located in oocytes and granulosa cells (cumulus cells) [58,59,60,61]; it would be of great interest to determine the intracellular signaling pathway by which melatonin promotes oocyte maturation and embryo development. While some reports have examined these intracellular processes of melatonin in oocytes [62,63,64,65], the details remain unclear. It has been reported that melatonin controls the expression of genes related to oocyte maturation including mitochondrial function [53,60,66,67], antioxidative enzymes [53,59,67,68], apoptosis [52,65,66,67,68,69], cumulus cell expansion [51,61,70], and oocyte maturation factors [61,67]. Furthermore, epigenetic mechanisms such as DNA methylation and histone acetylation have also been reported [59,66,71]. Further study may identify how these epigenetic mechanisms contribute to oocyte maturation by regulating the expression of some specific genes in oocyte and granulosa cells. In addition to a direct antioxidative action, it is essential to elucidate the detailed mechanisms of melatonin on oocytes and granulosa cells (cumulus cells) that involve both the membrane and nuclear receptors.

ROS play crucial roles in reproductive functions such as the ovulatory process. However, in excess, they may adversely affect oocytes in the form of oxidative stress which would cause infertility. Melatonin existing in follicles may possibly protect oocytes from ROS with its antioxidant activity and its involvement in oocyte maturation, fertilization, and embryo development. The trials of clinical application of melatonin for infertile women have reported improved outcomes of ART. It should be noted that melatonin supplementation could become a new treatment for improving oocyte quality and it may benefit women who suffer from infertility. 

## 6. Reduced Fertility Associated with Ovarian Aging

Age is the critical factor that has a major effect on fertility. Infertility caused by ovarian aging is the most important challenge in reproductive medicine. ART success decreases with increased age, with a sudden decline in the pregnancy rate and elevation in the miscarriage rate observed from 35 years of age onwards [72]. Ovarian aging has two problems and includes both reduced oocyte number and reduced oocyte quality. At birth, females are born with 1–2 million oocytes in their ovaries. However, they never produce any new oocytes thereafter and many are continuously being lost. It has been shown that at puberty, the number already is reduced to 100,000–300,000 oocytes, and that by the late 30s, particularly after 37 years of age, the number of oocytes rapidly decreases. 

As women grow older, the quality of each oocyte also is reduced. Oocytes are subjected to various types of damage in an age-dependent manner over the long period of several decades, with dysfunction noted in organelles such as mitochondria and nuclei. This is perceived as poor oocyte quality, and is a major cause of frequent unbalanced chromosome segregation in the first meiotic division accompanied by impaired fertilization [73]. Although the mechanisms that cause a decline oocyte quality have not been fully elucidated, it is generally accepted that oxidative stress caused by ROS contributes to age-induced dysfunction of oocyte mitochondria and nuclei. It has also been frequently reported that reduced antioxidative function caused by aging are associated with lower oocyte quality [74,75]. While attempts have been made to prevent the lower function and to improve the impaired quality of oocytes, at present no effective method has been established.

## 7. Anti-Aging Effects of Melatonin

Melatonin has drawn attention as an anti-aging molecule [2,9,23,24]. The life expectancy of mice treated with melatonin is reportedly prolonged, and the life expectancy of mice with the pineal gland removed is shortened. Therefore, there is a possibility that melatonin has the ability to delay ovarian aging. Tamura et al. examined the protective effect of melatonin on ovarian aging using mice [76]. Female mice were administered melatonin (drinking water containing melatonin) over a period from 10 to 43 weeks of age, after which oocytes were retrieved from follicles and IVF was performed. In 43-week-old control mice, there were fewer follicles of all developmental stages within the ovaries (primordial follicles, primary follicles, secondary follicles, and antral follicles). In the melatonin-treated animals, however, there were more follicles remaining compared to the number in the control animals. Also in the melatonin group, there were more ovulated oocytes, and the age-related decline in the number of oocytes was reduced. The results of IVF indicated that the number of fertilized oocytes and the number of blastocysts declined with age, but were maintained in the melatonin-treated animals. Thus, melatonin also appears to reduce the decline in oocyte quality. The authors also analyzed changes in ovarian gene expression using microarray. The expression of 77 genes decreased with age and were elevated as a result of melatonin treatment. Among these genes, approximately half (40 genes) were involved in ribosomal function. Furthermore, upon performing pathway analysis, the authors extracted eukaryotic initiation factor 2 (eIF2) signaling, which maintains the accuracy of protein synthesis (translation) (Figure 4). This was interpreted to mean that melatonin maintains ribosomal function, accuracies of gene translation, and protein synthesis, thereby slowing the processes of aging. On pathway analysis, the growth arrest and DNA-damage-inducible 45 (GADD45) signaling involved in DNA repair and checkpoint functions were predominantly noted, suggesting that melatonin enhances the mechanism underlying DNA damage repair (Figure 4). Melatonin also suppresses autophagy-related protein (light-chain 3a: LC3a; light-chain 3b: LC3b) by enhancing intracellular pathways, including eIF2, GADD45, and alternative reading frame (ARF) pathways. Furthermore, the network analysis found that melatonin had a stimulatory effect on antioxidative mechanisms. Moreover, the telomere length, which typically decreases during aging, and expressions of the sirtuin longevity genes (SIRT1, SIRT3) were significantly higher in the melatonin-treated animals compared to the control mice. Collectively, the results suggest that, through various mechanisms, melatonin reduces at least some aging processes in the ovaries and oocytes (Figure 4).

Differences in the time of initiating melatonin treatment might result in different anti-aging effects on the ovaries. Therefore, melatonin treatment was started from 23 or 33 weeks of age in mice, and the results of IVF outcomes at 43 weeks of age were analyzed according to the previous report [76] (Figure 5). In the group in which treatment was initiated at 23 weeks, the number of ovulated oocytes (8.5 ± 2.2, C23 (control group); 16.8 ± 3.0, M23 (melatonin group)), fertilization rate (32.3%, C23; 59.5%, M23), and blastocyst rate (17.6%, C23; 39.2%, M23) were all significantly higher in the melatonin-supplemented mice than in control animals (Figure 5). These results show that melatonin treatment that began at 23 weeks delayed aging of the ovaries. Conversely, in the group that started at 33 weeks, the number of ovulated oocytes (9.6 ± 1.8, C33; 9.4 ± 2.7, M33), fertilization rate (30.2%, C33; 37.6%, M33), and blastocyst rate (22.9%, C33; 36.4%, M33) showed no significant difference as a result of melatonin treatment compared to controls (Figure 5). Melatonin was not found to be beneficial if melatonin treatment began at 33 weeks. The implication is that, to reduce aging of the ovaries, melatonin supplementation should be initiated well in advance of reproductive deterioration.

In another study where melatonin was given to mice for 6–12 months, it was reported that it inhibited the age-related reduction in the number of follicles, litter sizes, and blastocyst rates while improving mitochondrial function, lowering ROS production, reducing oxidative stress, increasing adenosine triphosphate (ATP) production, lowering apoptosis, and elevating antioxidative enzymes, thus demonstrating that melatonin can reduce ovarian aging [77]. A recent study that investigated the effect of melatonin in delaying ovarian aging also reported on the importance of intracellular signaling (MT1/MAPK pathway) via melatonin receptors in oocytes (MT1) to modulate the age-related changes [78]. 

The mechanisms related to the anti-ovarian aging effects of melatonin are clearly not yet well defined. The clinical application of melatonin for the treatment of humans to improve ovarian physiology should have high priority. Many women suffer from the infertility due to ovarian aging. If long-term melatonin treatment prevents ovarian aging as represented by a decline in the number and quality of oocytes, they would have more oocytes of better quality when they undergo IVF-ET program. Melatonin treatment may make a major contribution to reproductive medicine to improve ART outcomes. There are currently no effective methods or established medications that prevent ovarian aging; the administration of melatonin may be a promising candidate for this purpose.

## Figures and Tables

**Figure 1 ijms-21-01135-f001:**
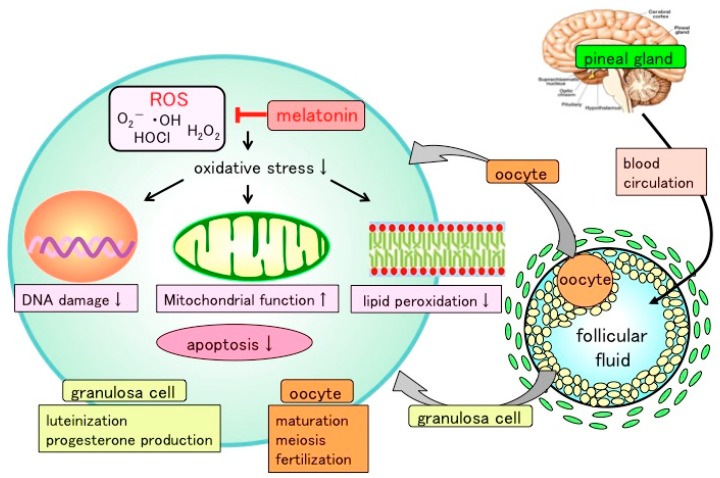
Presumed action of melatonin in ovarian follicle. Melatonin, secreted by pineal gland, is taken up into the follicular fluid from the blood. Reactive oxygen species (ROS) produced within the follicles, especially during the ovulation process, are scavenged by melatonin. Excess amounts of ROS may be involved in oxidative stress of oocyte and granulosa cells. Melatonin reduces the oxidative-stress-induced DNA damage, mitochondrial dysfunction, lipid peroxidation, and apoptosis of granulosa cells, showing that melatonin protects these cells by reducing free radical damage of cellular components including nuclei, mitochondria, and plasma membranes. The balance between ROS and antioxidants (melatonin) within the follicle may be critical for oocyte maturation, meiosis, and luteinization of granulosa cells.

**Figure 2 ijms-21-01135-f002:**
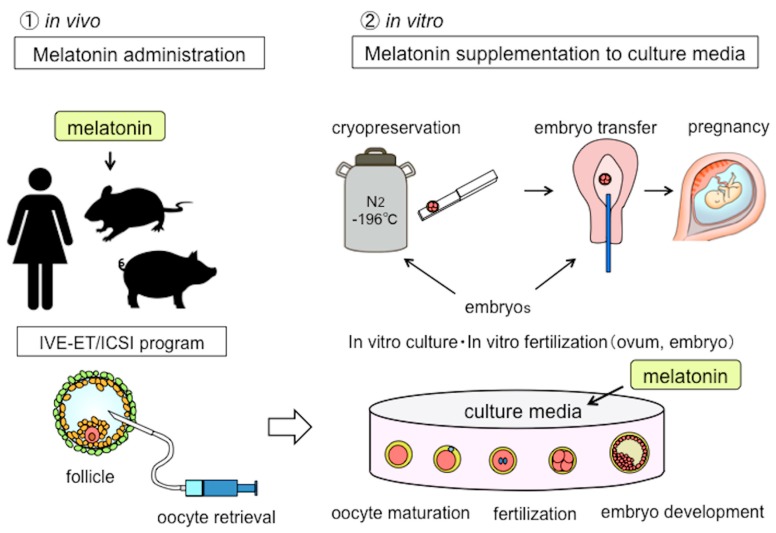
The potential applications of melatonin in human reproduction. Since the application of melatonin has antioxidant effects in reproductive medicine, there are two possibilities. One is that in vivo melatonin administration to patients before ovulation may improve the oocyte quality. Another possibility is melatonin supplementation added to in vitro culture media to enhance oocyte maturation, fertilization, and embryonic development. IVF-ET: in vitro fertilization and embryo transfer, ICSI: intra-cytoplasmic sperm injection.

**Figure 3 ijms-21-01135-f003:**
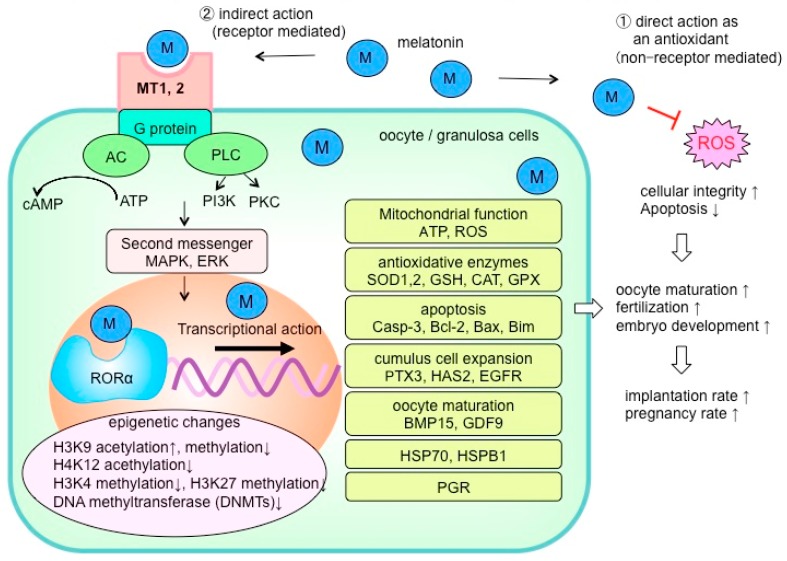
The reported mechanisms by which melatonin improves oocyte quality. The actions of melatonin are to be expected as a direct antioxidant effect to alleviate reactive oxygen species (ROS) and oxidative stress. Another indirect action of melatonin via cell membrane receptors (MT1, MT2) and nuclear receptor (RORα) also is considered to be very important for oocyte maturation and embryonic development. It is reported that antioxidant enzyme activity in oocytes, the expression of apoptosis-related factors, expression of genes involved in oocyte maturation and embryonic development, and epigenome changes such as DNA methylation and histone acetylation can be regulated by melatonin supplementation. ROS: reactive oxygen species; AC: adenylyl cyclase; PLC: phospholipase C; ATP: adenosine triphosphate; PI3K: phosphatidylinositol-3 kinase; PKC: protein kinase C; MAPK: mitogen-activated protein kinase; ERK: extracellular signal-regulated kinase; SOD: superoxide dismutase; GSH: glutathione; CAT: catalase; GPX: glutathione peroxidase; Casp: caspase; Bcl-2: B-cell lymphoma-2; Bax: Bcl-2-accociated X protein; Bim: Bcl-2 interacting mediator of cell death; PTX3: pentraxin-3; HAS2: hyaluronan synthase 2; EGFR: epidermal growth factor receptors; BMP: bone morphogenic protein; GDF: growth differentiation factor; HSP: heat shock protein; PGR: progesterone receptor.

**Figure 4 ijms-21-01135-f004:**
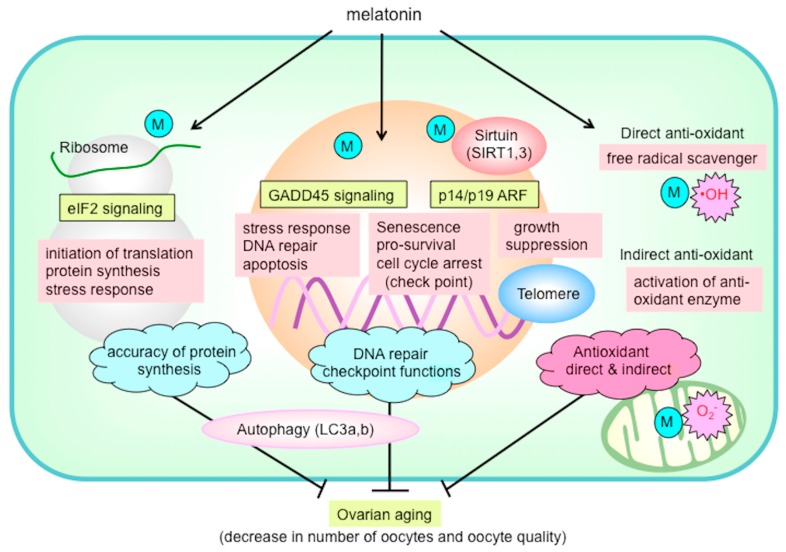
The possible mechanism of melatonin to prevent ovarian aging. Melatonin is likely to reduce ovarian oxidative stress not only by its direct action as a free radical scavenger but also by its indirect action of enhancing the antioxidant enzyme activity. Melatonin enhances eukaryotic initiation factor 2 (eIF2) signaling, which is essential for translation initiation and protein synthesis in ribosomes, and growth arrest and DNA-damage-inducible 45 (GADD45) signaling, which is involved in DNA repair and checkpoint functions. Melatonin also suppresses autophagy-related protein (light-chain 3a, 3b: LC3a, LC3b) by enhancing intracellular pathways including eIF2, GADD45, and alternative reading frame (ARF) pathways. The mRNA expression of sirtuin longevity genes (*SIRT1, SIRT3*) and telomere length were also enhanced due to melatonin treatment. Melatonin delays ovarian aging by multiple mechanisms including antioxidant action, DNA repair, maintaining telomeres, *SIRT* family activity, ribosome function, and autophagy. M: melatonin; O_2_^•−^: superoxide anion; OH: hydroxyl radical

**Figure 5 ijms-21-01135-f005:**
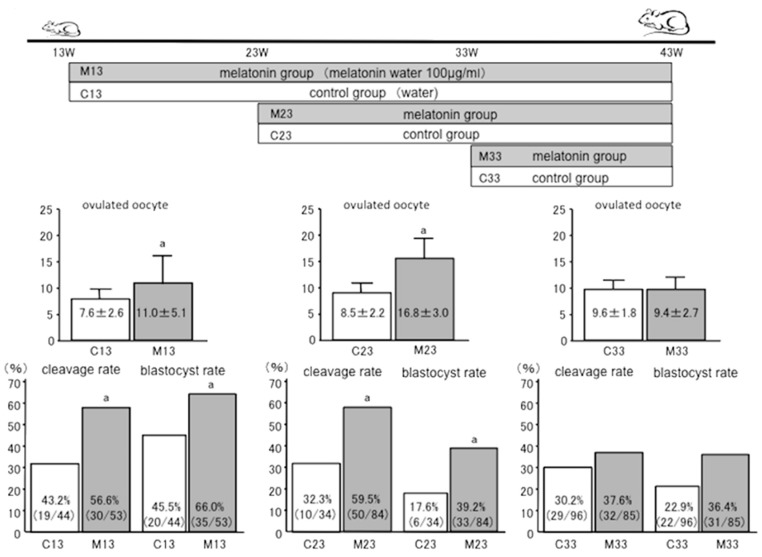
The anti-aging effects of melatonin on ovaries depends on the age of initiation of melatonin treatment. Melatonin treatment was started from 23 weeks (M23 weeks; melatonin group: M23, control group: C23) or 33 weeks (M33 weeks; melatonin group: M33, control group: C33) of age in mice, and the results of IVF outcomes at 43 weeks of age were analyzed. In the 23-week group, the number of ovulated oocytes (8.5 ± 2.2, C23; 16.8 ± 3.0, M23), fertilization rate (32.3%, C23; 59.5%, M23), and blastocyst rate (17.6%, C23; 39.2%, M23) were all significantly higher in the melatonin group than the control animals, and melatonin was found to have an anti-aging effect on the ovaries. On the other hand, in the 33-week group, the number of ovulated oocytes (9.6 ± 1.8, C33; 9.4 ± 2.7, M33), fertilization rate (30.2%, C33; 37.6%, M33), and blastocyst rate (22.9%, C33; 36.4%, M33) showed no significant difference between the melatonin treated and control animals.

**Table 1 ijms-21-01135-t001:** Effects of melatonin on assisted reproductive technology in humans (in vivo study). M: melatonin; C: control; IVF: IVF: in vitro fertilization; ICSI: intra-cytoplasmic sperm injection; 8-OHdG: 8-hydroxy-2′-deoxyguanosine; FF: follicular fluid; TAC: total antioxidant capacity.

Patients	Number	Technique	Melatonin Treatment	Result	Mechanisms	Year	Author/Reference
infertile women	115 (56M/59C)	IVF-ET	3mg/day orally	improved fertilization rate	reduced 8-OHdG in FF increased M in FF	2008	Tamura [6]
infertile women	60 (30M/30C)	IVF-ET	3mg/day orally	increased mature oocyte increased good quality embryos		2011	Eryilmaz [47]
infertile women	85 (40M/45C)	IVF-ET	3mg/day orally	increased mature oocyte increased good quality embryos		2012	Batioglu [46]
infertile women	97 (97M/97C)	IVF, ICSI	3mg/day orally	improved fertilization rate increased good quality embryos		2014	Nishihara [49]
infertile women diminished ovarian reserve	66 (32M/24C)	IVF, ICSI	3mg/day orally	increased mature oocyte increased good quality embryos		2017	Jahromi [48]
infertile women	30 (10C/10M, 10M)	IVF, ICSI	3 or 6mg/day orally	increased no of oocyte retrieved increased good quality embryos	increased M, TAC in FF decreased 8-OHdG in FF	2019	Espino [50]

**Table 2 ijms-21-01135-t002:** Effects of melatonin on assisted reproductive technology under in vitro conditions. COC: cumulus oocyte complex; IVM: in vitro maturation; M II: metaphase II; ROS: reactive oxygen species; MT1, MT2: melatonin membrane receptors; BMP: bone morphogenic protein; PTX3: pentraxin-3; HAS2: hyaluronan synthase 2; EGFR: epidermal growth factor receptors; GSH: glutathione; ATP: adenosine triphosphate; GDF: growth differentiation factor; SOD: superoxide dismutase; GPX: glutathione peroxidase; Bcl-2: B-cell lymphoma-2; GSX: reduced glutathione; OCT4: octamer-binding transcription factor 4; H2AX: histone H2 family member X; H3K4me: methylation of lysine4 on histone H3; H4K27me: methylation of lysine27 on histone H3; CAT: catalase; HSP: heat shock protein; MTNR1A: melatonin receptor 1A; MT: melatonin receptor; 4P-PDOT: 4-phenyl-2-propionamidotetralin; BimEL: Bcl-2 interacting mediator of cell death extra-long; ERK: extracellular signal-regulated kinase; H3K9: histone H3 lysine 9.

Animal	Design	Melatonin Treatment	Result	Year	Author/Reference
mouse	vitro COCs	10^−6^ M	Cumulus expansion, M-Ⅱ ↑ ROS, Acetyla on level of H4k12 ↓	2017	Keshavarzi Somayeh [51]
mouse	vitro, IVM, implantation	10^−7^ M	blastocyst rate, hatching blastocyst rate and blastocyst cell number ↑ pregnancy rate and birth rate↑, (ROS) production and cellular apoptosis ↓	2017	Tian Xiuzhi [69]
sheep	vitro, IVM	10^−7^ M	rates of nuclear maturation, cumulus cells expansion, cleavage, and blastocyst ↑ MT1 and MT2 were expressed in oocytes, cumulus cells, and granulosa cells BMP15, PTX3, HAS2, EGFR ↑, cAMP level ↓, cGMP ↑	2017	Tian Xiuzhi [65]
bovine	vitro, IVM	10^−9^ M	ROS↓, GSH↑,mitochondrial normal distribution increase ATP level upregulated ATPase 6, BMP-15, GDF-9, SOD-1, Gpx-4, and Bcl-2 downregulated apoptotic gene expression of caspase-3.	2017	Yang Minghui [68]
porcine	vitro IVM COCs	10^−7^, 10^−6^, 10^−5^ M	oocyte quality, embryo development ↑ ROS generation, apoptosis, and DNA damage ↓, GSX, OCT4, H2AX	2018	Lin Tao [52]
bovine	vitro, IVM	10^−9^ M	G1 blastocyst ↑, cell number ↑, apoptotic cell ↓ glutathione content, mitochondrial membrane potential ↑ antioxidant gene (SOD2) heat shock protein (HSPB1) ↑	2018	Marques TC [53]
porcine	vitro prolonged culture	10^−3^ M	blastocyst rate↑ methylation at H3K4me2 and H3K27me2 ↓ imprinted gene NNAT ↓	2018	Nie Junyu [71]
bovine	vitro, IVM	10^−9^ M	blastocyst, total cell number ↑, apoptotic cell ↓ ROS ↓, GSH ↑ caspase-3 ↓, BCL-2, XIAP, CAT, HSP70 ↑	2018	Pang Yunwei [66]
Goat	vitro, IVM	10^−9^ M, 10^−12^ M	M-II stage, blastocyst ↑, GSH ↑, MTNR1A in cumulus cell and oocytes DNA methyltransferases (DNMTs) global DNA methylation ↓	2018	Saeedabadi Saghar [63]
mouse	vitro, IVM	10 μM	fertilization rate ↑ hyaluronan synthase-2 (HAS2) and Progesterone receptor (PGR) ↑	2018	Ezzati Maryam [70]
porcine	vitro COCs	10^−9^ M	blastocyst, cell number, cumulus expansion ↑, apoptosis ↓ MT2 was expressed in both oocytes and cumulus cells M effects were abolished when either luzindole or 4P-PDOT (MT antagonist)	2018	Lee Sanghoon [62]
porcine	vitro COCs	10^−9^ M	pro-apoptotic protein BimEL, ERK-mediated phosphorylation M only promoted the ubiquitination of phosphorylated BimEL M action was independent of its receptor and its antioxidant properties	2018	Wang Yingzheng [67]
bovine	vitro cloned embryo	10^−9^ M	cloned embryo development ↑ oxidative stress, apoptosis, mitochondria, chromosome alignment epigenetic modifications, H3K9 acetylation ↑, H3K9 methylation ↓	2019	An Quanli [56]

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
