# Peer review of "Importance of Melatonin in Assisted Reproductive Technology and Ovarian Aging"

_ijms, 2020, doi:10.3390/ijms21031135_

Round 1

Reviewer 1 Report

The work is very interesting and it comprehensively exlores the role of melatonin in the biology of ovary development and aging and in mammals fertility.

I have only some minor comments listed below, to be addressed prior to publication

1-A lot of imprecisions are noticed throught the texts. Authors should perform english editing

2-I’m not sure about the term “discharged” in the abstract

3-In the introduction section, line 66, is not the rythm that is involved in many deseases, but alterations of these rythms. Add some review or articles about the correlatino between alteration of cyrcadian rythms, melatonin and the described desease.

4-Line 78, add references about melatonina s a better antioxidant respect to vitamine C, E, mannitol, glutathione.

5-Page 2, line 82, substitute “avtions” with “actions”

6-Page 3 , line 30 “in patients given melatonin” substitute with “in patients who received “melatonin

7-Simplify lines 136-150. Too many experimental details, describe only the take home message of the work from Tanabe et al. Same comment for the description of the work of Tamura et al. Explain the meaning of each acronymous i.e PMSG, ART.

8-Page6, line 238, replace the sentence"it has reported" with “it has been reported”

9-Page 7, 8, the paragraph about ovarian aging contains the repetition of the same concepts more than once. Simplify

10-Page 8, line 293 the sentence “This is consistent with melatonin’s ability to delay ovarian aging” is wrong in that context. Prolonged life is not consistent with delaied ovarian aging. Authors should rephrase the sentence

Author Response

Reviewer 1 comments

The work is very interesting and it comprehensively exlores the role of melatonin in the biology of ovary development and aging and in mammals fertility.

I have only some minor comments listed below, to be addressed prior to publication

Response to Reviewer 1

Thank you for your valuable suggestions.  We have made every attempt to make the suggested changes.

1-A lot of imprecisions are noticed throught the texts. Authors should perform english editing

    We corrected the mistakes carefully throughout the manuscript.

2-I’m not sure about the term “discharged” in the abstract

    We changed “discharged” to “secreted”.

3-In the introduction section, line 66, is not the rythm that is involved in many deseases, but alterations of these rythms. Add some review or articles about the correlatino between alteration of cyrcadian rythms, melatonin and the described desease.

    We added following appropriate references.

Cipolla-Neto J, Amaral FG, Afeche SC, Tan DX and Reiter RJ (2014) Melatonin, energy metabolism, and obesity: a review. J Pineal Res 56:371-81. doi: 10.1111/jpi.12137 Favero G, Franceschetti L, Bonomini F, Rodella LF and Rezzani R (2017) Melatonin as an Anti-Inflammatory Agent Modulating Inflammasome Activation. Int J Endocrinol 2017:1835195. doi: 10.1155/2017/1835195 Ma N, Zhang J, Reiter RJ and Ma X (2019) Melatonin mediates mucosal immune cells, microbial metabolism, and rhythm crosstalk: A therapeutic target to reduce intestinal inflammation. Med Res Rev. doi: 10.1002/med.21628 Moradkhani F, Moloudizargari M, Fallah M, Asghari N, Heidari Khoei H and Asghari MH (2020) Immunoregulatory role of melatonin in cancer. J Cell Physiol 235:745-757. doi: 10.1002/jcp.29036

4-Line 78, add references about melatonina s a better antioxidant respect to vitamine C, E, mannitol, glutathione.

    We added following reference.

Reiter RJ (1995) Functional pleiotropy of the neurohormone melatonin: antioxidant protection and neuroendocrine regulation. Front Neuroendocrinol 16:383-415. doi: 10.1006/frne.1995.1014

5-Page 2, line 82, substitute “avtions” with “actions”

    We changed “avtions” to “actions”.

6-Page 3 , line 30 “in patients given melatonin” substitute with “in patients who received “melatonin

     We changed “in patients given melatonin” to “in patients who received melatonin”.

7-Simplify lines 136-150. Too many experimental details, describe only the take home message of the work from Tanabe et al. Same comment for the description of the work of Tamura et al. Explain the meaning of each acronymous i.e PMSG, ART.

    We simplified the description of works of Tanabe et al. and Tamura et al.

8-Page6, line 238, replace the sentence "it has reported" with “it has been reported”

    We changed “it has reported” to “it has been reported”.

9-Page 7, 8, the paragraph about ovarian aging contains the repetition of the same concepts more than once. Simplify

     We simplified the paragraph about ovarian aging. We deleted the scentence “Age-related reductions in the number of growing follicles and number of retrieved oocytes are an important cause of poor fertility in women of advanced age.”.

10-Page 8, line 293 the sentence “This is consistent with melatonin’s ability to delay ovarian aging” is wrong in that context. Prolonged life is not consistent with delaied ovarian aging. Authors should rephrase the sentence

     We changed the sentence “This is consistent with melatonin’s ability to delay ovarian aging.” to “Therefore, there is a possibility that melatonin has ability to delay ovarian aging.”.

Reviewer 2 Report

This paper is reviewing the importance of melatonin in assisted reproductive technology as well as in ovarian aging. The possibilities of improved outcome of assisted reproductive technology like in vitro fertilization and embryo transfer (IVF-ET) by clinical application of melatonin are also discussed.

Melatonin is potent antioxidant and acts as a free radical scavenger and as such, melatonin can serve as an anti-aging molecule.

The manuscript is well written, informative and deal with interesting and important topics.

Minor points: there are few typographical errors in the text, and a sentence in page 4, line148 “These results document while ROS damage DNA, mitochondria, and cell membranes of granulosa cells, while melatonin prevents this mutilation thereby protecting granulosa cells.” is a little bit difficult to follow, so please describe it better.

I would recommend English proofreading which will improve the quality of the paper.

Figure 3 has explanation, but I would recommend that authors additionally explain the figure in the text, which will give better understanding of mechanism by which melatonin improves oocyte quality.

I believe that it will be better if the authors provide a separately a paragraph with conclusion.

Author Response

Reviewer 2 comments

This paper is reviewing the importance of melatonin in assisted reproductive technology as well as in ovarian aging. The possibilities of improved outcome of assisted reproductive technology like in vitro fertilization and embryo transfer (IVF-ET) by clinical application of melatonin are also discussed.

Melatonin is potent antioxidant and acts as a free radical scavenger and as such, melatonin can serve as an anti-aging molecule.

The manuscript is well written, informative and deal with interesting and important topics.

Response to Reviewer 2

Thank you for your valuable suggestions.  We have made every attempt to make the suggested changes.

Minor points: there are few typographical errors in the text, and a sentence in page 4, line148 “These results document while ROS damage DNA, mitochondria, and cell membranes of granulosa cells, while melatonin prevents this mutilation thereby protecting granulosa cells.” is a little bit difficult to follow, so please describe it better.

    We simplified the description of works of Tanabe et al.

I would recommend English proofreading which will improve the quality of the paper.

    Thank you for your advice. We corrected the mistakes carefully throughout the manuscript.

Figure 3 has explanation, but I would recommend that authors additionally explain the figure in the text, which will give better understanding of mechanism by which melatonin improves oocyte quality.

    We added explanations of Figure 3 and made some changes in the section of “Oocyte maturation, embryo development and melatonin”.

I believe that it will be better if the authors provide a separately a paragraph with conclusion.

    We added conclusion both infertility section and ovarian aging section separately.